# FAST TRAINING DATASET ATTRIBUTION VIA IN-CONTEXT LEARNING

## ABSTRACT

We investigate the use of in-context learning and prompt engineering to estimate the contributions of training data in the outputs of instruction-tuned large language models (LLMs). We propose two novel approaches: (1) a similarity-based approach that measures the difference between LLM outputs with and without provided context, and (2) a mixture distribution model approach that frames the problem of identifying contribution scores as a matrix factorization task. Our empirical comparison demonstrates that the mixture model approach is more robust to retrieval noise in in-context learning, providing a more reliable estimation of data contributions.

## 1 INTRODUCTION

Training Data Attribution (TDA) refers to the task of quantifying the contributions of different data sources on the output of a model (Park et al., 2023; Nguyen et al., 2023). This task is essential for debugging the curating corpora processes for training and for improving the training of neural networks Xia et al. (2024); Qin et al. (2025). Understanding the contribution of data sources allows us to assess the monetary value of proprietary training data, which is crucial for fair compensation and data management (Ghorbani & Zou, 2019; Nohyun et al., 2022; Choe et al., 2024). Unlike retraining- or gradient-based approaches, our methods require no model internals and instead exploit in-context learning behavior, making them directly applicable to black-box LLMs.

Existing TDA methods fall mainly into two categories: retraining-based methods and influence function-based methods, as detailed in recent surveys (Hammoudeh & Lowd, 2024; Worledge et al., 2024). Retraining approaches such as those of (Feldman & Zhang, 2020; Ghorbani & Zou, 2019) involve retraining the model without the target data source. However, this method is computationally expensive. The influence function approaches (Koh & Liang, 2017; Pruthi et al., 2020; Chen et al., 2021; Park et al., 2023), relax the need for full retraining by requiring only a few gradient calculations with respect to the data. Despite their efficiency, these methods rely on a linear approximation of the neural network around the target data point, which can be inaccurate. Critically, the influence function approaches compute the attribution score for a dataset as a linear function (usually an average or sum) of the attribution scores for each data point in the dataset (Hammoudeh & Lowd, 2024; Park et al., 2023). This approach fails to provide a holistic view of the contributions of an entire dataset to the model's output. Additionally, both methods require access to the internals of LLMs, which is not feasible for some popular models. A related technique, *Machine Unlearning* (Ginart et al., 2019; Sekhari et al., 2021) is still expensive to obtain the contribution scores.

We explore the use of in-context learning and prompt engineering to estimate the contributions of each dataset as a whole in the outputs of instruction-tuned LLMs. We propose two approaches: (1) A similarity-based approach, which posits that providing a dataset as context to an LLM trained on that dataset changes its output less compared to when the LLM was not trained on the dataset. (2) A mixture distribution model approach, where we model the behavior of LLMs using a new mixture distribution. This approach transforms the problem of identifying contribution scores into a matrix factorization problem, which we solve using the alternating projected least squares method. Both approaches utilize Retrieval Augmented Generation (RAG) (Lewis et al., 2020) to accommodate large data sources.

In the experiments, we evaluated four instruction-tuned LLMs: Mistral 7B (Jiang et al., 2023), Bloomz (Le Scao et al., 2023), Microsoft/Phi-3-mini (Abdin et al., 2024) and GPT 4.0 (Achiam et al.,

2023) on a set of binary Q&A datasets, BoolQ (Clark et al., 2019). In addition to the widely used BoolQ dataset, we create two new datasets: FakeQ, a synthetically modified version of BoolQ with altered queries and contexts, and a novel dataset constructed from Olympic 2024 Paris information, which serves as a realistic dataset that none of the LLMs have encountered during training. We evaluate our methods on these three datasets: BoolQ (likely seen), FakeQ (synthetic unseen but structurally similar), and Olympic2024 (constructed after model training and guaranteed unseen).

Finally, to ensure the reliability of our proposed contribution estimation metrics, we fine-tune these LLMs in the Olympic 2024 dataset under varying conditions, such as different learning rates, and evaluate the consistency of the metrics. Once validated, the metrics are further used to assess and rank popular unlearning techniques. Our contributions are two novel black-box methods for Training Data Attribution (TDA): (i) the Shapley Context Method (SCM), a similarity-based residual measure, and (ii) Context Mixture Factorization (CMF), a mixture-model formulation that is robust to retrieval noise. We evaluate these against Trak (Park et al., 2023) as a baseline. To mitigate concerns about ground truth, we validate our metrics using controlled fine-tuning on the Olympic2024 dataset, where attribution values increase monotonically with known exposure.

## 2 METHODOLOGY

An LLM processes knowledge from different sources. Our goal is to examine different prompts and see if we can uncover the sources of this knowledge.

In our setting, we have tuples in the format of question, context, and outcome: $(q, c, y)$. Our LLM outputs $M(q|c) = p(y|q, c)$. When we do not use any context, we denote $c = \emptyset$. Our goal is to quantify the contributions of the training datasets $D_1, \ldots, D_n$ in $p(y|q, c)$. We assume that we have a query set $Q = \{q_1, \ldots, q_m\}$. For simplicity of notation, without loss of generality, we describe the methods for binary outcome $y \in \{0, 1\}$.

We assume that we have $k, k = 1, \ldots, K$, relevant datasets about a topic and we want to quantify their contributions in the generation of the outputs by our LLM.

### 2.1 THE NON-PARAMETRIC APPROACH: THE SHAPLEY CONTEXT METHOD (SCM)

The key idea of this approach is that if an LLM uses the information from the $k$th dataset, providing the $k$th dataset as a context will not change the output much.

This assumption follows prior work in data Shapley (Ghorbani & Zou, 2019). and influence functions (Koh & Liang, 2017), where residual changes in predictions are treated as marginal contributions. SCM captures the direct effect of adding dataset context, making it sensitive to immediate residual changes but less robust to retrieval noise. We later validate this empirically.

We define the following similarity scores:

$$s_k = \text{sim}(y, y|c_k), \tag{1}$$

where $c_k$ is the context from the $k$th dataset.

Usually, the desired information can be found in multiple data sets (Ghorbani & Zou, 2019). To consider the impact of datasets in presence of other datasets, we define the following scores to be used in the Shapley formula (Shapley, 1953):

$$s_S = \text{sim}(y, y|c_S).$$

Intuitively, this measures the marginal gain in similarity when dataset $D_k$ is added.

The Shapley values are computed as follows:

$$\phi_k = \sum_{S \subseteq \{D_1, \ldots, D_K\} \setminus \{D_k\}} C_{S,K}(s_{S \cup \{D_k\}} - s_S), \tag{2}$$

where $C_{S,K} = |S|!(K - |S| - 1)!/K!$ are the normalization constants. This formula finds the residual increase in the similarity by including $D_k$, when we already have included another set $S \subseteq \{D_1, \ldots, D_K\} \setminus \{D_k\}$. The following Algorithm describes the details of our Shapley Context Method (SCM).

## 2.2 THE SEMI-PARAMETRIC APPROACH: CONTEXT MIXTURE FACTORIZATION (CMF)

Mixture modeling has long been used to capture latent source contributions in machine learning (Nguyen et al., 2023). Here we adapt this idea for dataset attribution: separating a base component ($\pi_0$) from dataset-specific components improves robustness to retrieval noise.

We propose a model to summarize the behavior of LLMs. Our model explicitly contains attribution scores and captures the entirety of the datasets used for its training. We use a mixture distribution approach, which defines:

$$p(y|q) = \pi_0 \widetilde{p}_0(y|q) + \sum_{k=1}^{K} \pi_k \widetilde{p}_k(y|q), \tag{3}$$

where $\widetilde{p}_0$ denotes a general-purpose language model and $\widetilde{p}_k$ denote the language models specialized on each of the relevant datasets $k = 1, \ldots, K$. The distributions $\widetilde{p}_k, k = 0, \ldots, K$, are latent, and we do not intend to explicitly estimate them. Intuitively, CMF is expected to outperform SCM in noisy RAG settings because $\pi_0$ explicitly accounts for background contributions, reducing variance in dataset-specific weights. This robustness also explains why SCM–CMF gaps vary across models: CMF separates background from dataset-specific influence, while SCM reflects only the marginal residual.

*Remark 1:* Given the modularity of LLM structures, this assumption is not fully realistic. However, this assumption provides a holistic view of the contributions of each dataset, captured by distributions $\widetilde{p}_k, k = 1, \ldots, K$. Thus, model (3) serves as a useful tool to statistically summarize the behavior of the LLM.

*Remark 2:* Model (3) can capture the scenarios where an LLM uses data from multiple sources, but does not model the scenarios where the LLM uses the interaction of data from multiple sources.

We model the impact of providing context from a dataset $k \in \{1, \ldots, K\}$ as an intervention in the probability distribution:

$$p(y|q, c_k) = \pi_0 \widetilde{p}_0(y|q) + (1 - \pi_0)\widetilde{p}_k(y|q). \tag{4}$$

The key assumption is that both Eq. (3) and (4) do not have context terms on the right-hand side quantities.

**Goal:** Our goal is to identify $\pi_k, k = 1, \ldots, K$. We want to do this without explicitly estimating $\widetilde{p}_k, k = 1, \ldots, K$.

**Formulating as a Matrix Factorization Problem.** For each of the $m$ queries, we perform $K + 1$ prompts (or the maximum $2^K$ prompts) and write the results in a linear equation as follows:

$$P = \Pi\widetilde{P}, \tag{5}$$

where $P \in [0, 1]^{(K+1) \times m}$, $\Pi \in [0, 1]^{(K+1) \times (K+1)}$, and $\widetilde{P} \in [0, 1]^{(K+1) \times m}$. We observe the quantity on the left-hand side, but none of the quantities in the right-hand side.

Eq. (5) expresses observed outputs as a mixture of latent dataset-specialized models, making attribution equivalent to estimating mixture weights.

This is a matrix factorization problem with a special structure. We assume that $\widetilde{p}_k(y|q)$ can be obtained by some clever prompts. We can make assumptions about $\widetilde{p}_k(y|q)$ that allow recovery of the mixture parameters of $\boldsymbol{\pi}$.

*Remark 3:* Instead of $K + 1$ prompts, we can have up to $2^K$ prompts. However, for the prompts that use multiple datasets, we need to assume the form of the resulting distribution, similar to Eq. (4). An alternative is to impose priors on $\boldsymbol{\pi}$ and $\widetilde{P}$ to improve identifiability. We will discuss the second approach in the next section.

**Alternating Projected Least Squares.** We can have multiple estimates for $\boldsymbol{\pi}$ from Eq. (5). We can resolve this issue by encouraging solutions that have lower variance. We achieve this by using two regularizers: an entropy regularizer for $\boldsymbol{\pi}$ to assume that the sources contribute equally and a regularizer that encourages $\widetilde{P}$ to be less informative.

$$\widehat{\boldsymbol{\pi}} = \arg\min_{\boldsymbol{\pi}} \min_{\widetilde{P}} \left\| P - \Pi\widetilde{P} \right\|_F^2 - \lambda_\pi H(\boldsymbol{\pi}) + \lambda_{\widetilde{P}}\|\widetilde{P} - 1/2\|_F^2, \tag{6}$$

$$\text{s.t.} \quad \boldsymbol{\pi} \succeq \mathbf{0}, \quad \mathbf{1}^\top \boldsymbol{\pi} = 1, \quad 0 \preceq \widetilde{P} \preceq 1.$$

where $\|\cdot\|_F$ and $H(\cdot)$ denote the Frobenius norm and Shannon's entropy. We use entropy regularization on $\boldsymbol{\pi}$ to encourage the null hypothesis of "equal contributions of all sources". Regularization of the Frobenius norm implies that, unless there is strong evidence, the outputs of the latent probabilities $\widetilde{P}$ should be $1/2$. Note that regularizers are vital for obtaining a non-trivial solution and in the absence of them, there are many solutions to the problem.

The problem in Eq. (6) is biconvex; ie, fixing $\boldsymbol{\pi}$ or $\widetilde{P}$, the problem is convex (Gorski et al., 2007). Thus, we solve it by the alternating least-squares method. We further assist in the regularization terms by randomly initializing $\widetilde{P}$ to be around $1/2$ and $\boldsymbol{\pi}$ to be around $1/(K+1)$. We can obtain the confidence intervals for both SCM and CMF by bootstrapping (Tibshirani & Efron, 1993).

## 3 IMPLEMENTATION

### 3.1 PROMPT ENGINEERING

For simplicity of evaluation and without loss of generality, we used Q&A datasets, where the answers are binary Yes/No. To instruct the LLMs to provide direct boolean responses, we used prompt engineering. Initially, we tested various prompts without explicitly instructing the model to answer with "Yes" or "No." Diverse examples used in this process are provided in Appendix A. Through iterative testing, we found that the responses improved when the model was explicitly instructed to provide a Boolean answer. This led to our final prompt:

**Prompt:** "Given the context below, answer the question that follows with only 'Yes', 'No', or 'I don't know' if the context is insufficient.
{question}? The answer to this question is "

Although this final prompt worked well for GPT-4, Bloomz, and Mistral 7B, generating straightforward "Yes," "No," or "I don't know" responses, it was harder to instruct Phi-3-mini. Even with the final prompt, Phi-3-mini often generated more text than just a simple boolean response.

Therefore, calculating similarities was straightforward for GPT-4, Bloomz, and Mistral 7B, but we had to devise another solution for Phi-3-mini. The embedding similarity API on GPT-4 was not precise enough as it did not focus primarily on the context of the generated response. To calculate the similarity for Phi-3-mini, we created a zero-shot classification layer (which takes 1000 characters) between the prediction and the result to measure similarity more accurately.

### 3.2 USING RAG

Given the limitations of LLM context windows, fitting entire datasets directly into the context is impractical. To address this, we used Retrieval Augmented Generation (RAG) (Lewis et al., 2020) to enhance context by retrieving relevant documents from databases before generating responses. The process involves splitting the documents into semantically relevant chunks using the *RecursiveCharacterTextSplitter* from the HuggingFace Transformers library, computing embeddings for all chunks with a model like *thenlper/gte-small*, and storing these embeddings in a vector database using FAISS (Facebook AI Similarity Search) Johnson et al. (2019). When a question is posed, it is embedded, and a similarity search is performed against the vector database to find the closest matching documents. These retrieved documents are then provided as context for the LLMs along with the original question, allowing the LLMs to generate responses augmented with additional context. We used a chunk size of 512 and a top-k value of 3, ensuring the context was trimmed to 2000 characters for conciseness. We study the effectiveness of RAG in the Appendix C. While we focus on BoolQ-style binary QA for tractability, our framework is general and can extend to multi-class or free-form generation. Future work will apply SCM and CMF to such outputs, building on case studies like OLMoTrace.

## 4 EXPERIMENTS

Simplified setup to demonstrate our methodology:

**Step 1: Task Selection** We use three datasets for our evaluation: (1) the BoolQ Q&A dataset (Clark et al., 2019), which consists of tuples in the form (question, relevant context, binary answer), representing a dataset which we treat as likely seen during pretraining; (2) the FakeQ dataset, constructed by altering the queries and contexts in BoolQ to ensure the dataset remains unseen by the LLMs; and (3) The Olympic 2024 dataset, a newly created dataset based on the Paris 2024 Olympics (detailed in Appendix B), is designed to simulate real-world scenarios with Yes/No questions and relevant contexts. The latter two datasets allow us to evaluate the attribution metrics on datasets to which the LLMs have not been exposed during pretraining. Using BoolQ (likely seen), FakeQ (unseen but structurally similar), and Olympic2024 (constructed post-training and guaranteed unseen) allows us to systematically test attribution under different exposure assumptions. These datasets were selected to control exposure. Although we cannot confirm pretraining details for closed models, the contrast between BoolQ (likely seen), FakeQ (synthetically unseen), and Olympic2024 (constructed post-training) provides a minimal sanity check against stylistic confounds.

**Step 2: LLM Selection** We examined four instruction-tuned LLMs: GPT-4 (1.76 trillion parameters), Bloomz (176 billion parameters), Mistral 7B (7.3 billion parameters), and Phi-3-mini (3.8 billion parameters). We report the accuracy of these LLMs on BoolQ in Table 9. Given that the dataset is binary, we prompted the LLMs to answer "Yes" or "No" to each question, or to say "I don't know" if they could not provide a definite response (see Section 3.1).

**Step 3: Alternative Dataset Collection** We collected five datasets on different topics. The corpora were sourced from a subset of the Wikipedia Field of Science dataset available on Hugging Face, specifically the fields of Chemistry, Natural Science, History and Archaeology, Biology, and Law. Each of these data sets contains more than a million samples in five categories. The five science domains, drawn from Wikipedia, were not intended as unseen data but as broad controls; their lower attribution compared to BoolQ highlights that general background knowledge contributes less to BoolQ-style QA than task-specific data.

**Step 4: Evaluation** First, we evaluated the methods on the BoolQ dataset, which is closely related to the questions asked, providing a baseline for how well the methods estimate attribution for data sources that align closely with the LLMs' pretraining. Successful methods should estimate a higher weight for BoolQ, *as a proxy for the relevant data* used during training.

To further test the robustness of the proposed methods, we created two new datasets: (1) FakeQ, derived by altering the queries and contexts in BoolQ to ensure it is entirely unseen by the LLMs, and (2) Olympic2024, a dataset based on the Paris 2024 Olympics, designed as a real-world dataset with binary Yes/No questions and relevant contexts that are guaranteed to be unseen by the LLMs (trained prior to 2024). These data sets allowed us to investigate how attribution metrics behave when the datasets have no prior exposure during LLM training.

To demonstrate the validity of the attribution metrics, we performed a series of experiments involving fine-tuning the LLMs on the Olympic2024 dataset with increasing learning rates or iterations. By fine-tuning the LLMs incrementally on a previously unseen dataset, we created a progression of models. For each fine-tuned model, we applied the contribution estimation methods and observed that the metrics increased monotonically, reflecting the LLMs' growing reliance on the dataset.

Finally, we extend our evaluation to machine unlearning algorithms, leveraging the established metrics to rank well-known unlearning methods based on their ability to effectively reduce the contribution of a target dataset. By applying attribution metrics to LLMs subjected to unlearning processes, we assessed whether the influence of the targeted dataset was effectively diminished.

### 4.1 RESULTS AND ANALYSIS

In general, SCM and CMF demonstrate their ability to effectively identify the most influential datasets, with CMF providing more robust attributions by accounting for noise and base contributions. Importantly, while attribution values for BoolQ may appear expected, the Olympic2024 experiments show the methods also capture signal in genuinely novel data, supporting their use beyond trivial cases.

We also performed an evaluation using Trak (Park et al., 2023) as a popular baseline. Trak provides a different perspective on attribution of data sets by scoring the impact of training data on model predictions. For our methods to "work," we expect: high attribution for BoolQ (seen/familiar), lower for FakeQ (synthetic unseen), and lowest for Olympic2024 (novel). Larger CMF–SCM gaps indicate stronger robustness to retrieval noise. We present the results for BoolQ, FakeQ, and Olympic2024 in Tables 1, 2, and 3.The Trak scores for Bloomz and Phi-3-mini in the BoolQ, FakeQ, and Olympic2024 datasets are shown in Tables 1, 2, and 3. The gap between CMF and SCM narrows on FakeQ and Olympic2024 because neither dataset was seen during training, reducing overlap. In contrast, BoolQ shows the largest gap, consistent with its alignment to pretraining data.

Table 1: Attribution values for the context on BoolQ

| Algorithm | Bloomz | GPT-4 | Mistral 7B | Phi-3 |
|---|---|---|---|---|
| SCM | 0.48 | 0.59 | 0.57 | 0.50 |
| CMF | **0.63** | **0.62** | **0.61** | **0.59** |
| Trak | 0.61 | – | – | 0.58 |

Table 2: Attribution values for the context on FakeQ

| Algorithm | Bloomz | GPT-4 | Mistral 7B | Phi-3 |
|---|---|---|---|---|
| SCM | 0.32 | 0.36 | 0.33 | 0.28 |
| CMF | **0.46** | **0.43** | **0.41** | **0.38** |
| Trak | 0.39 | – | – | 0.35 |

Table 3: Attribution values for the context on Olympic2024

| Algorithm | Bloomz | GPT-4 | Mistral 7B | Phi-3 |
|---|---|---|---|---|
| SCM | 0.08 | 0.11 | 0.09 | 0.07 |
| CMF | **0.16** | **0.14** | **0.12** | **0.10** |
| Trak | 0.12 | – | – | 0.09 |

**Detailed Analysis of Attribution Coefficients** Tables 4 and 5 show the attribution results obtained by the SCM and CMF algorithms for the BoolQ data set. Both algorithms successfully identify the BoolQ dataset as the most influential dataset. This is because the BoolQ context is more directly related to the questions. Chemistry, Natural Science, History and Archaeology, Biology, and Law have lower $\phi_k$ values, showing that while they contribute to the context, their impact is less significant compared to BoolQ. Note that in CMF, we need to calculate $\frac{\pi_{\text{BoolQ}}}{1-\pi_{\text{Base}}}$ to directly compare it with $\phi_{\text{BoolQ}}$ estimated by SCM. This shows that CMF assigns higher attribution values than SCM due to its robustness in accounting for noise in retrieval-augmented generation (RAG) systems.

Table 4: Shapley Values ($\phi_k$) using SCM Algorithm.

| Metric | Bloomz | GPT-4 | Mistral 7B | Phi-3 |
|---|---|---|---|---|
| $\phi_{\text{BoolQ}}$ | 0.48 | 0.59 | 0.57 | 0.50 |
| $\phi_{\text{Chemistry}}$ | 0.10 | 0.08 | 0.09 | 0.10 |
| $\phi_{\text{Natural Sci}}$ | 0.12 | 0.09 | 0.10 | 0.11 |
| $\phi_{\text{History}}$ | 0.11 | 0.10 | 0.10 | 0.11 |
| $\phi_{\text{Biology}}$ | 0.10 | 0.07 | 0.08 | 0.10 |
| $\phi_{\text{Law}}$ | 0.09 | 0.07 | 0.08 | 0.08 |

Table 5: $\pi$ and $\frac{\pi_{\text{BoolQ}}}{1-\pi_{\text{Base}}}$ values using the CMF algorithm.

| Metric | Bloomz | GPT-4 | Mistral 7B | Phi-3 |
|---|---|---|---|---|
| $\pi_{\text{Base}}$ | 0.05 | 0.08 | 0.06 | 0.05 |
| $\pi_{\text{BoolQ}}$ | 0.60 | 0.62 | 0.61 | 0.59 |
| $\pi_{\text{Chemistry}}$ | 0.09 | 0.07 | 0.08 | 0.09 |
| $\pi_{\text{Natural Sci}}$ | 0.07 | 0.08 | 0.07 | 0.06 |
| $\pi_{\text{History}}$ | 0.08 | 0.10 | 0.09 | 0.07 |
| $\pi_{\text{Biology}}$ | 0.06 | 0.06 | 0.05 | 0.05 |
| $\pi_{\text{Law}}$ | 0.05 | 0.10 | 0.08 | 0.06 |
| $\frac{\pi_{\text{BoolQ}}}{1-\pi_{\text{Base}}}$ | 0.63 | 0.67 | 0.65 | 0.62 |

Table 6: TRAK Scores for Different Sources using Phi-3 and Bloomz models. Positive scores indicate datasets that contribute positively to the model's output, while negative scores indicate a lesser or inverse influence.

| Dataset | Phi-3 | Bloomz |
|---|---|---|
| BoolQ | 0.58 | 0.61 |
| Chemistry | -0.08 | -0.10 |
| Natural Science | 0.20 | 0.22 |
| History | 0.18 | 0.19 |
| Biology | -0.05 | -0.06 |
| Law | 0.07 | 0.09 |

Across the three datasets (see Tables 10–13 in Appendix D), CMF consistently assigns higher attribution values than SCM. This is because CMF explicitly accounts for background model contributions ($\pi_{\text{Base}}$) and dataset-specific contributions ($\pi_k$), making it more robust to noise and improving attribution granularity. The difference between CMF and SCM is most pronounced for datasets with strong alignment with the pre-training data of the model, such as BoolQ, where CMF captures a clearer and stronger attribution signal. Since BoolQ contains questions and contexts similar to those likely encountered during model training, CMF detects these relationships with greater sensitivity.

For FakeQ and Olympic2024, both unseen during pre-training, the gap between CMF and SCM narrows. This is expected as neither data set has direct overlap with pre-training data, leading to lower attribution values across the board. However, CMF still assigns slightly higher attributions compared to SCM, particularly for FakeQ. This suggests that while FakeQ is designed to be unseen, it retains enough linguistic patterns and contextual structures resembling BoolQ for CMF to register a weak but measurable connection. In contrast, Olympic2024 shows the lowest attribution values, reflecting its novel and domain-specific nature. This trend underscores the ability of the CMF to differentiate datasets not only based on direct exposure but also through latent associations in linguistic or contextual patterns, making it a more reliable metric for evaluating dataset contributions.

These findings validate the superior sensitivity of CMF in identifying relevant data influences, even for datasets with no explicit pre-training overlap. At the same time, they illustrate that both methods converge to lower attribution values when applied to data sets entirely outside of the prior knowledge of the model, confirming the robustness of SCM and CMF in distinguishing between the data sets seen and the novel ones.

For BoolQ, Trak identified it as the most influential dataset, which aligns well with our methods. However, our CMF approach provides a more detailed and accurate attribution of dataset contributions, particularly in quantifying the base model's influence and managing the noise inherent in retrieval-augmented generation (RAG) systems. CMF consistently assigns higher attribution values compared to SCM and Trak, reflecting its robustness in capturing the alignment between BoolQ and the model training data.

For FakeQ, Trak shows lower attribution scores compared to BoolQ, as expected for an unseen dataset. CMF again outperforms SCM and Trak by effectively taking advantage of the syntactic similarity of FakeQ to BoolQ, capturing its partial alignment with training data. SCM also performs reasonably well, but is less sensitive to subtle contributions, and Trak provides scores comparable to SCM, though it lacks the granularity CMF offers.

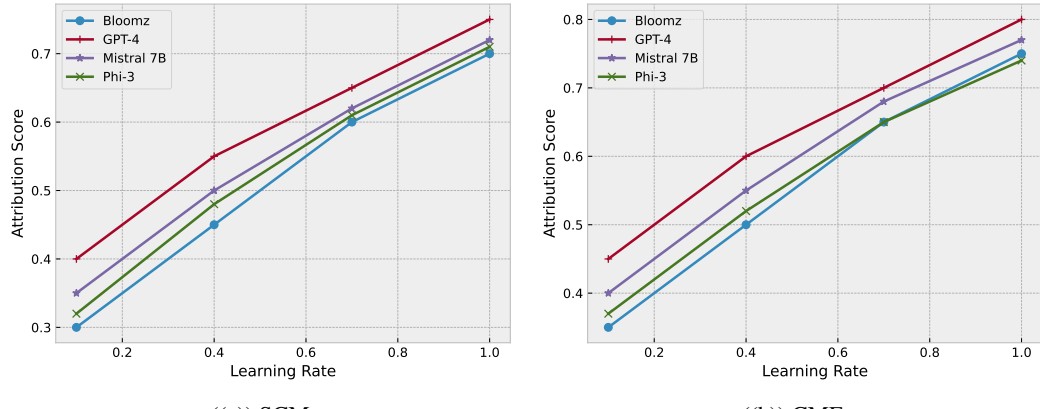

((a)) SCM                                 ((b)) CMF

Figure 1: (left) SCM Attribution Values vs. Learning Rate for Olympic2024 Dataset: Attribution values increase with fine-tuning. (right) CMF Attribution Values vs. Learning Rate for Olympic2024 Dataset: CMF shows higher attribution values, reflecting its robustness during fine-tuning.

Table 7: SCM and CMF Attribution Scores for the Olympic2024 Dataset.

| Unlearning Method | LLM | Before Finetuning | | After Finetuning | | After Unlearning | |
|---|---|---|---|---|---|---|---|
| | | SCM | CMF | SCM | CMF | SCM | CMF |
| Gradient Ascent | Bloomz | 0.08 | 0.16 | 0.75 | 0.85 | 0.25 | 0.34 |
| | Phi-3 | 0.07 | 0.10 | 0.72 | 0.82 | 0.28 | 0.37 |
| Fine-tuning with Random Labels | Bloomz | 0.08 | 0.16 | 0.75 | 0.85 | 0.22 | 0.31 |
| | Phi-3 | 0.07 | 0.10 | 0.72 | 0.82 | 0.24 | 0.33 |
| Unlearning with Adversarial Samples | Bloomz | 0.08 | 0.16 | 0.75 | 0.85 | 0.30 | 0.41 |
| | Phi-3 | 0.07 | 0.10 | 0.72 | 0.82 | 0.33 | 0.42 |

For Olympic2024, being entirely novel and unrelated to training data, all methods report significantly lower attribution values. CMF continues to demonstrate superior performance by reflecting even minimal data-set contributions while maintaining a clear distinction between seen and unseen data. SCM and Trak exhibit a smaller gap between Olympic2024 and FakeQ, indicating their limited ability to fully capture the novelty of datasets.

**Case Study: Evaluation of Unlearning Methods**   We applied attribution metrics to assess the effectiveness of machine unlearning methods. As shown in Table 7, three unlearning methods: Gradient Ascent (Golatkar et al., 2020; Liu et al., 2024), Fine-tuning with Random Labels (Golatkar et al., 2020), and Unlearning with Adversarial Samples (Cha et al., 2024) were evaluated based on their ability to reduce the influence of the Olympic2024 dataset on Bloomz and Phi-3 models.

The results indicate that Unlearning with Adversarial Samples consistently outperforms the other methods, achieving the highest reduction in attribution values for both SCM and CMF metrics. The ability of this method to target specific data points for unlearning is reflected in the reduced values for $\frac{\pi_{\text{Olympic}}}{1-\pi_{\text{Base}}}$. In contrast, Gradient Ascent and Fine-tuning with Random Labels achieve moderate reductions, with Gradient Ascent slightly outperforming Random Labels in most cases.

**Runtime Comparison**   The CMF algorithm is faster than the SCM algorithm as it requires fewer queries with shorter context sizes. Using an AWS EC2 G6 instance (g6.16xlarge), the total runtime for CMF, involving 7 runs, ranges from 77 to 94 minutes for all LLMs. In contrast, the SCM method, which requires $2^5$ runs, results in a total runtime of 352 to 384 minutes. The runtimes of both algorithms are dominated by the RAG search time. This substantial reduction in run-time demonstrates the efficiency of the CMF method, making it more suitable for scenarios demanding both accuracy and computational efficiency.

For Trak, the runtime is significantly higher due to its high memory requirements. Trak requires about 20 GB of GPU memory for a model with 1 million parameters. Scaling this to larger models, Trak's memory requirements become impractical for large LLMs with modest computing resources. Running Trak on our LLMs would necessitate approximately 600 GB of GPU memory and significantly more computational time, making CMF and SCM more feasible for our use case.

**Validation of Attribution Metrics through Fine-Tuning**  To establish the reliability of the attribution metrics, we performed a fine-tuning experiment on the Olympic2024 dataset. Fine-tuning was performed incrementally, with the learning rate increasing progressively. Figures 1(a) and 1(b) demonstrate that the attribution values increase monotonically with fine-tuning, confirming that the metrics effectively capture the growing dependence of LLM on the data set. For both SCM and CMF, the Olympic2024 attribution values increase with fine-tuning, but CMF shows higher attribution values due to its ability to handle noise more robustly. These results validate the ability of the proposed metrics to quantify the influence of fine-tuned data on LLMs.

The incremental increase in attribution values also reflects a nonlinear growth pattern, as the effects of fine-tuning diminish at higher learning rates, resulting in a plateau. This plateau effect is more prominent in SCM as it lacks the noise handling capabilities of CMF. This observation demonstrates the practical applicability of these metrics in scenarios where data influence evolves over time as a result of fine-tuning.

**Deep Dive into RAG Noise Effect**  We computed the mean similarities ($s_S$) and residuals ($r_{k,S} = s_{S \cup \{D_k\}} - s_S$) for the BoolQ datasets in all LLMs, as shown in Table 8. These metrics provide a nuanced understanding of how data sets influence model outputs.

For the BoolQ dataset, the high negative residual for Bloomz (-0.32) indicates that adding the BoolQ context significantly influences the model output. This substantial change highlights the alignment of the data set with the pre-existing knowledge of the model, as seen in the high similarity score ($s_S = 0.86$). In contrast, GPT-4's low residual (-0.03) and high similarity score ($s_S = 0.76$) suggest that it has been preexposed to similar data during training, resulting in minimal performance changes when BoolQ context is added. Mistral 7B, with a positive residual (0.06), demonstrates a strong reliance on the added BoolQ context, suggesting that it benefits greatly from this additional information. Similarly, Phi-3's small positive residual (0.03) indicates partial exposure to similar data but with room for improvement when additional context is provided. We present the results on FakeQ and Olympics2024 data in Appendix E. Together, these findings confirm that our metrics capture both strong pretraining overlap (BoolQ) and novel data (Olympic2024), with CMF better isolating background noise.

Table 8: Mean similarities $s_S$ and residuals $r_{k,S}$ for different LLMs (with standard deviations).

| **LLM** | $s_S$ | $s_{S \cup \{D_k\}}$ | $r_{k,S}$ |
|---|---|---|---|
| **Bloomz** | 0.86 (0.03) | 0.54 (0.02) | -0.32 (0.02) |
| **GPT-4** | 0.76 (0.02) | 0.72 (0.02) | -0.03 (0.02) |
| **Mistral7B** | 0.73 (0.03) | 0.69 (0.03) | 0.06 (0.02) |
| **Phi-3** | 0.60 (0.03) | 0.63 (0.02) | 0.03 (0.02) |

## 5  CONCLUSION AND DISCUSSION

Our results demonstrate that both proposed algorithms can successfully attribute LLM outputs to relevant training data. In particular, BoolQ serves as a proxy for "seen" knowledge: GPT-4 showed minimal change when BoolQ context was added, suggesting prior exposure, while Bloomz exhibited a large residual, indicating stronger reliance. CMF further disentangles dataset-specific and base model contributions, offering robustness to retrieval noise and lower computational cost compared to SCM. Fine-tuning experiments on Olympic2024 provide a controlled ground truth, confirming that attribution values increase predictably with known exposure.

Looking ahead, we see several natural directions. First, attribution should be scaled to much larger corpora, for example by batched prompting, adaptive sampling, or streaming updates in CMF. Second, extending beyond binary QA to free-form and multi-class outputs will require new signals, such as token-level or perplexity-based metrics. Third, retrieval design itself (e.g., chunk size, reranking, or long-context strategies) merits study, as it shapes the stability of attribution. Finally, our methods can be combined with complementary approaches such as string-based attribution and memorization audits, and evaluated on datasets with verified training exposure. Together, these directions point toward more comprehensive and practical frameworks for training data attribution in black-box LLMs.

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

## A PROMPTS

**General Question Prompt:** "Read the context provided and answer the following question: {question}"

**Contextual Understanding Prompt:** "Based on the information in the context, what can you conclude about the following question? {question}"

**Summarization Prompt:** "After considering the context below, summarize your answer to this question: {question}"

**Opinion-Based Prompt:** "Given the details in the context, what is your opinion on the following question: {question}"

**Detail Extraction Prompt:** "Extract relevant information from the context to answer this question: {question}"

**Fact-Checking Prompt:** "Using the context provided, verify the accuracy of the following statement: {question}"

## B OLYMPIC 2024 DATASET

The Olympic 2024 dataset was constructed by sourcing textual snippets from publicly available Kaggle datasets Kaggle (2024) related to the Paris 2024 Olympics. Since no structured QA dataset existed for this event, question-answer pairs were manually generated for each context snippet to create a

well-defined evaluation framework. For each retrieved context, three binary (Yes/No) question-answer pairs were created: one for training, one for validation, and one for testing. The training set was used to fine-tune the models, allowing them to incorporate Olympic-related information. The validation set was used to evaluate attribution metrics, ensuring that the methods assessed data influence without direct exposure during training. The test set was reserved for evaluating generalization after fine-tuning and unlearning experiments. This design follows the structure of BoolQ, making it easier to analyze how fine-tuning impacts attribution and how well unlearning methods reduce the model's reliance on the dataset.

## C  EFFECTIVENESS OF RAG

To evaluate the effectiveness of context provision using RAG, we designed an experiment to measure the accuracy of various LLMs when answering questions from the BoolQ dataset. The experiment compared the models' performance across different scenarios: without any context, with only the BoolQ context, with contexts from five other datasets, and with all datasets combined. The results are summarized in Table 9.

Table 9: Accuracy of LLMs with Different Contexts.

| Context | Bloomz | GPT-4 | Mistral 7B | Phi-3 |
|---|---|---|---|---|
| No Context | 0.43 | 0.73 | 0.68 | 0.70 |
| BoolQ as RAG | 0.74 | 0.87 | 0.85 | 0.82 |
| Five Datasets Only | 0.35 | 0.64 | 0.60 | 0.45 |
| All Data + BoolQ | 0.73 | 0.84 | 0.82 | 0.83 |

The baseline setting (No Context) reveals inherent differences in the LLMs' capabilities. GPT-4 has the highest baseline accuracy at 0.73, followed by Phi-3 at 0.70 and Mistral 7B at 0.68, indicating robust pretraining for these models. Bloomz shows lower accuracy at 0.43, highlighting its dependency on contextual data.

When the BoolQ context is provided using RAG, all models show significant accuracy improvements, with GPT-4 reaching 0.87, Mistral 7B at 0.85, and Phi-3 at 0.82. Bloomz also improves to 0.74, though it remains lower than the others. Providing context from five datasets (excluding BoolQ) leads to accuracy drops for all models, with Bloomz at 0.35, GPT-4 at 0.64, Mistral 7B at 0.60, and Phi-3 at 0.45. This indicates that less relevant data are less effective in understanding BoolQ queries.

Combining all data sets with the BoolQ context results in slight decreases in accuracy for GPT-4 (0.84) and Mistral 7B (0.82), suggesting that additional data sets introduce noise. Bloomz and Phi-3 show minimal changes, indicating that additional data do not significantly impact their performance once the BoolQ context is included. These results emphasize the importance of relevant contextual information in improving LLM performance, with GPT-4 consistently outperforming other models due to its extensive training.

## D  ADDITIONAL DETAILED RESULTS

Tables 10 and 11 display the attribution results for the FakeQ dataset, which was constructed by altering the queries and contexts in BoolQ to ensure that it is unseen by the LLMs. **SCM Attribution** ($\phi_{\textbf{FakeQ}}$)**:** The attribution values for FakeQ are noticeably lower than those for BoolQ. For example, $\phi_{\text{FakeQ}}$ ranges from 0.28 to 0.36 in LLM, compared to 0.48 to 0.59 for $\phi_{\text{BoolQ}}$. This reflects the lack of prior exposure to FakeQ in the training data, leading to a reduced alignment with the preexisting knowledge of the model.

**CMF Attribution** ($\pi_{\textbf{FakeQ}}$)**:**  CMF assigns higher scores than SCM, with $\frac{\pi_{\text{FakeQ}}}{1-\pi_{\text{Base}}}$ ranging from 0.48 to 0.53 in LLM. This increase indicates the sensitivity of the CMF to the structural similarities between FakeQ and BoolQ. Although FakeQ is unseen, its construction retains the semantic patterns of BoolQ, allowing the models to leverage these structural similarities.

Tables 12 and 13 present the attribution results for the Olympic2024 dataset, which is entirely unseen and constructed to simulate real-world scenarios. The findings here are markedly different:

**SCM Attribution ($\phi_{\text{Olympic2024}}$):** The Olympic2024 attribution values are significantly lower than both BoolQ and FakeQ. For example, $\phi_{\text{Olympic2024}}$ ranges from 0.07 to 0.11 in LLMs. This is expected since Olympic2024 is entirely unrelated to the models' pre-training data, and its context does not align with the questions posed.

**CMF Attribution ($\pi_{\text{Olympic2024}}$):** CMF similarly assigns lower attribution scores to Olympic2024 compared to BoolQ and FakeQ, with $\frac{\pi_{\text{Olympic2024}}}{1-\pi_{\text{Base}}}$ ranging from 0.15 to 0.20. However, the CMF values remain slightly higher than SCM, demonstrating its ability to account for small signal contributions even in unseen datasets. The behavior of the Olympic2024 attribution metrics highlights their reliability in distinguishing between datasets that are seen (BoolQ), partially similar (FakeQ), and entirely novel (Olympic2024). The low residuals for Olympic2024, particularly for models such as GPT-4, suggest minimal influence from prior training data, further validating the robustness of the attribution methods.

| Metric | Bloomz | GPT-4 | Mistral 7B | Phi-3 |
|---|---|---|---|---|
| $\phi_{\text{FakeQ}}$ | 0.32 | 0.36 | 0.33 | 0.28 |
| $\phi_{\text{Chemistry}}$ | 0.08 | 0.06 | 0.07 | 0.08 |
| $\phi_{\text{Natural Sci}}$ | 0.09 | 0.07 | 0.08 | 0.09 |
| $\phi_{\text{History}}$ | 0.08 | 0.07 | 0.08 | 0.08 |
| $\phi_{\text{Biology}}$ | 0.07 | 0.05 | 0.06 | 0.07 |
| $\phi_{\text{Law}}$ | 0.06 | 0.05 | 0.06 | 0.06 |

Table 10: Shapley Values ($\phi_k$) using SCM Algorithm on FakeQ Dataset.

Table 11: $\pi$ and $\frac{\pi_{\text{FakeQ}}}{1-\pi_{\text{Base}}}$ values using the CMF algorithm.

| Metric | Bloomz | GPT-4 | Mistral 7B | Phi-3 |
|---|---|---|---|---|
| $\pi_{\text{Base}}$ | 0.06 | 0.09 | 0.07 | 0.06 |
| $\pi_{\text{FakeQ}}$ | 0.45 | 0.48 | 0.47 | 0.42 |
| $\pi_{\text{Chemistry}}$ | 0.10 | 0.07 | 0.08 | 0.09 |
| $\pi_{\text{Natural Sci}}$ | 0.08 | 0.07 | 0.07 | 0.08 |
| $\pi_{\text{History}}$ | 0.09 | 0.08 | 0.08 | 0.08 |
| $\pi_{\text{Biology}}$ | 0.07 | 0.06 | 0.06 | 0.06 |
| $\pi_{\text{Law}}$ | 0.06 | 0.06 | 0.06 | 0.06 |
| $\frac{\pi_{\text{FakeQ}}}{1-\pi_{\text{Base}}}$ | 0.49 | 0.53 | 0.51 | 0.48 |

| Metric | Bloomz | GPT-4 | Mistral 7B | Phi-3 |
|---|---|---|---|---|
| $\phi_{\text{Olympic2024}}$ | 0.08 | 0.11 | 0.09 | 0.07 |
| $\phi_{\text{Chemistry}}$ | 0.04 | 0.03 | 0.04 | 0.05 |
| $\phi_{\text{Natural Sci}}$ | 0.05 | 0.04 | 0.04 | 0.05 |
| $\phi_{\text{History}}$ | 0.05 | 0.04 | 0.05 | 0.05 |
| $\phi_{\text{Biology}}$ | 0.04 | 0.03 | 0.03 | 0.04 |
| $\phi_{\text{Law}}$ | 0.03 | 0.02 | 0.03 | 0.03 |

Table 12: Shapley Values ($\phi_k$) using SCM Algorithm on Olympic2024 Dataset.

Table 13: $\pi$ and $\frac{\pi_{\text{Olympic2024}}}{1-\pi_{\text{Base}}}$ values using the CMF algorithm.

| Metric | Bloomz | GPT-4 | Mistral 7B | Phi-3 |
|---|---|---|---|---|
| $\pi_{\text{Base}}$ | 0.08 | 0.11 | 0.09 | 0.08 |
| $\pi_{\text{Olympic2024}}$ | 0.15 | 0.18 | 0.16 | 0.14 |
| $\pi_{\text{Chemistry}}$ | 0.07 | 0.05 | 0.06 | 0.06 |
| $\pi_{\text{Natural Sci}}$ | 0.06 | 0.05 | 0.06 | 0.06 |
| $\pi_{\text{History}}$ | 0.07 | 0.06 | 0.06 | 0.06 |
| $\pi_{\text{Biology}}$ | 0.06 | 0.05 | 0.05 | 0.05 |
| $\pi_{\text{Law}}$ | 0.05 | 0.05 | 0.05 | 0.05 |
| $\frac{\pi_{\text{Olympic2024}}}{1-\pi_{\text{Base}}}$ | 0.16 | 0.20 | 0.18 | 0.15 |

# E    EXTENDED DEEP DIVE INTO RAG NOISE EFFECT

For the FakeQ dataset, the results reveal its controlled construction and structural similarity to BoolQ. Bloomz exhibits a moderately negative residual (-0.16) and a slightly reduced similarity score ($s_S = 0.78$), indicating that although FakeQ is not identical to BoolQ, its design allows the model to relate to it effectively. GPT-4 maintains a low residual (-0.03), reinforcing its robustness in handling data sets that resemble those encountered during training. Mistral 7B and Phi-3 show small positive residuals (0.03 and 0.04, respectively), suggesting that these models benefit from the added FakeQ context while exhibiting a less direct alignment compared to BoolQ.

For the Olympic2024 dataset, the results underscore its novelty. Bloomz has a less negative residual (-0.13) and a reduced similarity score ($s_S = 0.72$), highlighting the limited alignment of this entirely novel data set with the model's pre-existing knowledge. GPT-4 continues to display a low residual (-0.03), showing its robustness even when handling unseen data. Mistral 7B and Phi-3 exhibit small positive residuals (0.03 and 0.05, respectively), indicating their reliance on added context to improve performance. The lower similarity scores across all models for Olympic2024 reflect the unique nature of the data set, distinguishing it from the other data sets.

Table 14: Mean similarities $s_S$ and residuals $r_{k,S}$ for FakeQ across different LLMs (with standard deviations).

| LLM | $s_S$ | $s_{S \cup \{D_k\}}$ | $r_{k,S}$ |
|---|---|---|---|
| **Bloomz** | 0.78 (0.04) | 0.62 (0.03) | -0.16 (0.03) |
| **GPT-4** | 0.69 (0.03) | 0.66 (0.02) | -0.03 (0.02) |
| **Mistral 7B** | 0.65 (0.04) | 0.68 (0.03) | 0.03 (0.03) |
| **Phi-3** | 0.58 (0.04) | 0.62 (0.03) | 0.04 (0.03) |

Table 15: Mean similarities $s_S$ and residuals $r_{k,S}$ for Olympic2024 across different LLMs (with standard deviations).

| LLM | $s_S$ | $s_{S \cup \{D_k\}}$ | $r_{k,S}$ |
|---|---|---|---|
| **Bloomz** | 0.72 (0.05) | 0.59 (0.04) | -0.13 (0.04) |
| **GPT-4** | 0.64 (0.03) | 0.61 (0.03) | -0.03 (0.03) |
| **Mistral 7B** | 0.60 (0.04) | 0.63 (0.03) | 0.03 (0.03) |
| **Phi-3** | 0.55 (0.05) | 0.60 (0.04) | 0.05 (0.04) |

