# OpenReview forum: "Fast Training Dataset Attribution via In-Context Learning"
_ICLR.cc/2026/Conference — Submitted to ICLR 2026_

### Official Review · Reviewer_SXoQ · 2025-10-17

**Soundness:** 2
**Presentation:** 3
**Contribution:** 3
**Rating:** 4
**Confidence:** 4

**Summary:**

The paper investigates an important and interesting topic: the contributions of training data to the outputs of instruction-tuned large language models (LLMs).
The authors propose two novel and easy-to-use methods: the Shapley Context Method and Context Mixture Factorization.
Experiments across various datasets show that the Context Mixture Factorization method consistently estimates data contributions.

**Strengths:**

1. The paper introduces two novel approaches (SCM and CMF) to address an important and interesting problem: estimating the contribution of training data to the outputs of instruction-tuned LLMs.

2. The paper is generally well-structured and easy to follow. The methodology is described in clear detail.

3. Comprehensive empirical analysis and ablation studies in the experiment section.

**Weaknesses:**

1. Missing comparison: The paper only compares SCM and CMF with the TRAK method proposed in 2023. The paper should compare SCM and CMF with the advanced and recent baseline methods from other studies. Such an analysis could offer valuable insights into the strengths and weaknesses of the proposed approaches.

2. Limited Improvement: Tables 1–3 show that the results of CMF and the baseline TRAK are quite similar, especially for the Phi-3 model. Moreover, SCM performs worse than TRAK. It would be helpful if the authors could further clarify the advantages of the proposed methods.

3. Lacks an analysis of its limitations. The success of the proposed methods relies on the assumptions shown in Remarks 1 and 2. It would be valuable for the authors to discuss specific scenarios in which these assumptions may not hold and the proposed methods are likely to fail.

4. Missing analysis on the selection strategy and the number of demonstrations used in ICL: It would be useful if the authors also studied how different selection strategies (e.g., EPR [1] and DPP [2]) and varying numbers of demonstrations (e.g., K = 4, 8, 16) affect the performance of the proposed methods.

5. Notation issues: The notation system of this paper can be further improved. For example, the definition of $c_S$ is unclear.

6. Citation issues: The author needs to correct the citation format of the whole paper. For example, in line 25, the citation format should be in the form of (Xia et al., 2024; Qin  et al., 2025).

7. The absence of code and generated datasets makes it difficult for the broader research community to reproduce the results claimed in the paper or verify the method's effectiveness on the tasks.

[1] Learning to retrieve prompts for in-context learning.

[2] Compositional exemplars for in-context learning

**Questions:**

See weaknesses

---

### Official Review · Reviewer_p8pQ · 2025-10-29

**Soundness:** 2
**Presentation:** 2
**Contribution:** 2
**Rating:** 4
**Confidence:** 3

**Summary:**

This paper addresses the problem of Training Data Attribution for large language models, specifically focusing on the black-box setting where internal model gradients and parameters are inaccessible. The authors note that existing TDA methods, such as retraining-based or influence function-based approaches, are often computationally expensive or require white-box access, making them unsuitable for many modern LLMs . To address this limitation, the paper proposes a framework that utilizes In-Context Learning and Retrieval-Augmented Generation. The core idea is to use RAG to provide dataset-specific context within a prompt and then measure the resulting change in the LLM's output to estimate the dataset's contribution.

**Strengths:**

It introduces the use of In-Context Learning and Retrieval-Augmented Generation as a non-intrusive "probe" to estimate dataset contributions in a black-box setting . This conceptual reframing of ICL/RAG from a performance-enhancement mechanism to a diagnostic tool for attribution is a new approach to the black-box TDA problem.

**Weaknesses:**

1) The paper's primary proposed method, CMF, is built on theoretically weak foundations. The core assumption (Eq. 4), which models how ICL context intervenes in the mixture distribution, appears to be entirely heuristic. The paper provides no theoretical derivation or empirical validation for this specific functional form; it seems to be a mathematical construct specifically chosen to make the subsequent matrix factorization (Eq. 5) tractable, rather than a principled model of the LLM's actual behavior during ICL.
2) The paper's experimental validation is entirely limited to binary question-answering Yes/No tasks. This is a low-dimensional, simplified task setting. It completely avoids the core challenge of TDA for modern LLMs: attributing free-form, generative text. The paper provides no evidence that its framework, which relies on the output probability distribution p(y|q,c), can be generalized to high-dimensional and complex generative tasks, which severely limits the claimed robustness and applicability of the findings.
3) The methodology's heavy reliance on a RAG system introduces a critical confounding variable. The resulting attribution score is an ambiguous measure that conflates the LLM's reliance on its training data with the quality and effectiveness of the RAG retriever. For instance, a low attribution score for the Olympic2024 dataset could be attributed to the RAG's embedding model failing to retrieve relevant context, rather than purely reflecting the LLM's lack of training exposure. The experiments do not adequately disentangle this ambiguity.

**Questions:**

1) Could the authors please elaborate on how the CMF framework, particularly the probability matrix P in Eq. 5, would be applied to free-form text generation tasks? What would the elements of P represent in that scenario sequence perplexity, specific token probabilities, or embedding similarities?
2) The proposed method tightly couples attribution scores with the performance of the RAG system, making it difficult to distinguish whether a low attribution score reflects the LLM's training or simply the RAG system's retrieval quality. The authors need to provide an ablation study on RAG quality. For example, if RAG quality for a seen dataset is intentionally degraded , how much does the CMF score decrease? Conversely, if an oracle context is provided for an unseen dataset, how much does its CMF score increase?

---

### Official Review · Reviewer_x1MU · 2025-10-31

**Soundness:** 2
**Presentation:** 1
**Contribution:** 2
**Rating:** 2
**Confidence:** 3

**Summary:**

The paper proposes two black-box, no-retraining methods to estimate how much specific training datasets contribute to a LLM's outputs. The SCM treats dataset context as in-context learning and measures residual changes in outputs, aggregating over context combinations with Shapley values. The CMF models observed outputs as a mixture of latent dataset-specialized components and recovers mixture weights by a constrained, regularized alternating least-squares factorization. Experiments on BoolQ, FakeQ, Olympic2024 set show that CMF is more robust to retrieval noise and that both metrics increase monotonically under controlled fine-tuning that raises exposure to Olympic2024. The metrics are further used to rank machine unlearning methods, where adversarial-sample unlearning yields the largest attribution drops, and CMF is substantially more time- and resource-efficient than SCM and far more practical than Trak.

**Strengths:**

1. The proposed method can be applied to black-box LLMs and does not require re-training.

2. Thoughtful dataset design covering likely seen, synthetic unseen but similar, and guaranteed unseen settings.

3. Demonstrated downstream use for evaluating unlearning methods and strong runtime efficiency for CMF.

**Weaknesses:**

1. The presentation of the proposed methodology is not smooth, which hinders comprehension. If the author intends to use mathematical symbols, it is necessary to clearly define them beforehand. Furthermore, many of the mathematical notations appear to be inaccurate. For example, denoting $p(y|q,c)$ as $M(q|c)$ is unconventional, and the notation used in Eq. (1) is imprecise, which may lead to misinterpretation of the author's intended meaning.

2. The rationale behind the described methods, SCM and CMF, does not appear to be well-founded. The exposition relies heavily on assumptions, and since the algorithms are presented without supporting evidence or intuitive justification, the methodology lacks persuasiveness.

3. As shown in Tables 1, 2, and 3, the baselines other than SCM and CMF appear insufficient. A more comprehensive comparison with additional baselines is necessary to empirically validate the effectiveness of the proposed methods.

4. The main experiments focus on binary yes or no outputs.

**Questions:**

1. Following Eq. (1), what exactly does $c_S$ denote?
2. Is Eq. (3) an assumption proposed by the authors? If so, please explain the rationale for this assumption in greater detail.
3. Please provide a more detailed derivation of the step from Eq. (3) to Eq. (4).

---

### Official Review · Reviewer_qMTm · 2025-10-31

**Soundness:** 1
**Presentation:** 1
**Contribution:** 1
**Rating:** 0
**Confidence:** 4

**Summary:**

This paper proposes two data attribution methods that can estimate the relative importance of different datasets to a model's prediction. Compared with other approaches such as Trak, these methods are claimed to be more efficient and accurate.

**Strengths:**

The potential to create a reliable data attribution method that is computationally efficient is good.

**Weaknesses:**

I found the explanation of the methods in this paper highly confusing. For 2.1, the most salient issues are:
* Line 074: "context" is not clearly defined. Based on the datasets used, it seems that "context" is a passage that is relevant for answering the question, but in other places "context" appears to refer to in-context demonstrations from a dataset.
* Line 092/Equation 1: While claimed to be a definition of $s_k$, this is not actually a definition as "sim" is also not defined. As a more minor point, I think this should say something like $sim(y_1, y_2 | c_k)$ instead of $sim(y, y | c_k)$ since the two y's should presumably be different.
* Line 098: Similarly, the definition of $s_S$ is unclear, both for the reason above ("sim" is not defined) and also because $S$ and $c_S$ are not defined. Given Equation 2, S probably denotes a set of datasets, so maybe $c_S$ is the concatenation of contexts from examples in $S$?
* Line 107: Points to an Algorithm that is not present

For 2.2, the most salient issues are:
* Line 136: The paper points out that the context terms are dropped from the RHS of equation 4, but the significance and reasonability of this assumption is not adequately discussed. This again relates to the unclear definition of "context". If "context" refers to passages required to answer the question, dropping the context-dependence seems like it would greatly alter the model's outputs. This may be less of an issue if "context" only refers to in-context examples, but still warrants justification.
* $\pi_0$ is assumed to be fixed, which seems like a big assumption. Intuitively, the prompt can do a lot to shift how much the model relies on this prior.
* Line 142: It is unclear what a "query" is. Do you make a separate query for each example? What is the result of a query--is it the model's probability assigned to the correct answer? The model's answer distribution?
* Line 144/Equation 5: The variables in this equation are not defined (only the dimensions are specified, not their semantics).
* Line 154/Remark 3: It's not clear why 2^K is the maximum number of prompts. 2^K is the number of subsets of datasets. But for each dataset, one can presumably form many prompts by choosing different examples from that dataset. How are datasets/subsets of datasets mapped to prompts?

Other presentation issues include:
* Line 195: It is claimed that "calculating similarities was straightforward" but it still needs to be specified how they were calculated.
* Line 198: The comment about 1000 characters was confusing.

This paper compares with Trak but does not compare with MAGIC, a more recent work from some of the same authors https://arxiv.org/pdf/2504.16430 . MAGIC claims to greatly outperform Trak.

Perhaps most importantly, the evaluation in this paper is extremely coarse-grained. The evaluation relies on testing whether the relative attribution strength of different datasets follows expected trends. While acceptable as a sanity check, this is not sufficient for evaluating a method like this. Papers like Trak or MAGIC compute much more fine-grained correlations between their attributions and empirical importance, and those best practices should be followed by this work.

Line 360: The paper claims that CMF can identify "Latent associations in linguistic or contextual patterns", but it seems that the evidence for this general claim is minimal.

Line 369: The paper claims that CMF "provides a more detailed and accurate attribution of dataset contributions", but I did not see any justification for why it is more accurate. The observed "details" could just be random fluctuations. In the same paragraph, the paper says that since "CMF consistently assigns higher attribution values" than other methods, this reflects "robustness in capturing alignment" but it's very unclear why a higher value is considered better. The goal of the attribution method should be to reflect the model's reliance on different data, not to produce a high value on the dataset *we* think should be most relied on.

**Questions:**

Minor points/suggestions for writing:
* Line 025: Should be \citep
* Line 075: The notation $M(q|c)$ seems strange to me, as it highly suggests that M assigns a probability to a question conditioned on a context. I think $M(q, c)$ is more appropriate.
* Line 079: You can just write "we have $k=1, \dotsc, K$"
* Line 250: Redundant explanation of Olympic2024, it was already described previously

---

### Official Review · Reviewer_4V1H · 2025-11-05

**Soundness:** 2
**Presentation:** 3
**Contribution:** 2
**Rating:** 4
**Confidence:** 3

**Summary:**

This paper introduces two dataset attribution methods that utilizes in-context learning. The first one measures the dataset contribution by the marginal change between with and without the dataset in the context. The second method uses a mixture model to estimate the contribution of each dataset. Empirically, the methods are evaluated on 4 models with 3 evaluation datasets.

**Strengths:**

- Using in-context learning to measure the dataset contribution is an interesting idea that has efficiency advantages.
- The authors conducted extensive experiments for empirical evaluation.
- The paper is easy to follow.

**Weaknesses:**

- The proposed methods have strong assumptions that are not explicitly validated.
- For the Shapley Context Model, a key (implicit) assumption is that adding the dataset to the context would achieve a similar effect as if the model is finetuned on this dataset. This could be validated empirically (e.g., by comparing the outputs of a base model + dataset context vs a model finetuned on the dataset).
- For the Context Matrix Factorization, the assumptions that in-context learning (and finetuning) can be modeled as mixture distributions are not convincing to me. These assumptions should also be empirically validated.

**Questions:**

Could the authors validate the assumptions in a more explicit way?

---

### Meta-Review · Area_Chair_kxZU · 2026-01-07

**Summary:**

This paper addresses the problem of training data attribution for large language models (LLMs) in the black-box setting, where retraining or gradient-based methods are infeasible. The authors propose two approaches: the Shapley Context Method (SCM) and Context Mixture Factorization (CMF), both leveraging in-context learning to estimate dataset contributions. The work is motivated by the need for efficient, practical attribution methods and is evaluated across several datasets and models.

**Reviewer Concerns:**

- Methodological assumptions: Across multiple reviews, a recurring concern is that both SCM and CMF rest on strong, largely unvalidated assumptions. For SCM, the assumption that adding a dataset to the context approximates fine-tuning effects is questionable. For CMF, the mixture model formulation appears heuristic and lacks theoretical justification.
- Notation and presentation: Reviewers consistently found the mathematical exposition confusing. Key terms (e.g., "context," "query," "sim") are not clearly defined, and several equations are imprecise or unconventional. Missing algorithms and unclear derivations further hinder comprehension.
- Evaluation limitations: The experiments are restricted to binary yes/no tasks, which oversimplify the attribution problem. The paper does not convincingly demonstrate generalizability to free-form generative tasks, which are central to modern LLMs.
- Baseline comparisons: The paper compares only against Trak, omitting more recent and relevant baselines such as MAGIC. This weakens the empirical validation of the proposed methods.
- Confounding factors: The reliance on retrieval-augmented generation introduces ambiguity, as attribution scores may conflate dataset reliance with retriever quality. No ablation studies are provided to disentangle these effects.

**Reviewer Scores:**

N/a

---

### Decision · Program_Chairs · 2026-01-26

Reject